# Production efficiency and change characteristics of China's apple industry in terms of planting scale

Yu Sun[1], Yonghua Lu[1]*, Zichun Wang[1], Mingyue Li[2]

**1** Department of Management, Qingdao Agricultural University, Qingdao, China, **2** School of Economics and Management, Beijing Forestry University, Beijing, China

* sylyac2006@163.com

**Data Availability Statement:** All relevant data are within the manuscript and its Supporting Information files.

**Funding:** This research was funded by the Shandong Province Modern Agricultural Industrial

## Abstract

The global population is rapidly increasing, the arable land area is losing in a large scale, and the water supply capacity is limited. Meanwhile, China is in a critical period of the transformation of apple industrial structure, and the improvement of apple production efficiency is an important way to increase farmers' output and income, moderate-scale operation is the inevitable trend in agricultural modernization. However, few studies have explored the production efficiency of the apple industry from the perspective of planting scale. In China, there are seven major apple-producing provinces: Shaanxi, Shandong, Gansu, Henan, Shanxi, Hebei, and Liaoning. Therefore, based on provincial panel data of the seven main apple-producing areas in China, this study used the Malmquist productivity index and data envelopment analysis to measure the efficiency level of the apple industry. At the same time, the threshold regression model was used to analyze the characteristics of the change in apple planting scale and production efficiency. The results showed that apple production efficiency in different regions of China exhibited regional differences and time series fluctuations. Apple planting scale had a "double" threshold effect, and the impact on apple production efficiency showed a "negative effect–positive effect" trend. Therefore, the suggestion is to appropriately adjust the scale of operation, take measures according to local conditions, promote the upgrading of apple production technology, and realize the integration of apple production and sales by using "Internet +."

## Introduction

Apple production is an important way to increase farmers' income and plays a major role in revitalizing the agricultural economy [1]. China is the world's largest producer and consumer of apples. In recent years, its apple industry has developed rapidly. Major adjustments have been made in planting system reform, technological upgrades, and organizational cultivation, and industrial development has gradually moved toward modernization. China has transformed from a major apple producer in the world to a powerful apple producer. However, the labor productivity of the small-scale agricultural production mode is low, and the added value

Technology System (SDAIT-01-13), the Project of Shandong Provincial Department of Education (No: M2020111), the high-level talent research start-up fund of Qingdao Agricultural University (No: 663 / 1116710), and Project of Qingdao Agricultural University: Research on the Training Mode of New Agricultural Talents in Agricultural Colleges and Universities (711/1120057).

**Competing interests:** There is no financial competition for this article.

of the economic benefit is not sufficient to make up for the increase in production costs [2, 3]. Moreover, the continuous increase in the labor force, land, fertilizer, and other means of production and chemical inputs restricts the high-quality and sustainable development of the apple industry [4, 5]. In addition, China's agricultural production environment not only varies greatly across the country but also varies within the same province in terms of agricultural resource endowment, technical level, and market efficiency [6, 7].Therefore, in the context of the rapid development of agricultural informatization, improving agricultural production efficiency has become a feasible measure to promote the green and healthy development of agriculture [8–10], and moderate-scale operation has also become an inevitable trend of agricultural modernization.

## Literature review

The influence of agricultural production scale on production efficiency has always been the focus of domestic and foreign scholars. First of all, research on the relationship between agricultural production scale and production efficiency. Some scholars believe that there is a positive relationship between them, believing there is a positive relationship between them [11, 12]. Some scholars have also explored this topic from the perspective of expanding farm scale, indicating that with the expansion of farm scale, the unit level decreases while agricultural production efficiency calculated by total factor productivity increases; that is, there is an "inverse relationship" in agricultural development [9, 12–15]. Other studies have found that the relationship between the expansion of production scale and agricultural production efficiency is not linear [16, 17] but presents an "inverted U-shaped" relationship [18–20]. For example, Luo and Yao [21] studied China's grain production from the angle of input-output and found that it has a low scale, only relies on scale expansion to increase production, has an unreasonable production element configuration, and is influenced by various resource endowments with regard to efficiency. Therefore, it is not possible to improve production efficiency simply by expanding the scale of production.

Secondly, some studies have confirmed that production scale expansion and production efficiency improvement result from the combined action of many family endowment characteristics (e.g., family scale, apple-planting experience) [16], natural environment factors (e.g., climate change, geographical position, and natural disasters) [19], facilities and management style (e.g., use of pesticides and irrigation, marketing systems, and organization). All of these are important factors that determine whether expanding production scale can improve production efficiency. Other studies have emphasized that the appropriate expansion of production scale is conducive to the improvement of agricultural production efficiency. However, there are also some uncertain factors, such as labor transfer and labor cost; it is necessary, therefore, to consider strengthening investment in the human capital of apple growers [22]. In addition, the knowledge level of workers plays a promoting role in agricultural production efficiency [11]. According to Ang [23], the overall lack of skill and knowledge contribution leads to the blocked improvement of production efficiency. All of the abovementioned studies affirmed the role of agricultural technology extension in improving production efficiency.

The studies discussed above laid a solid foundation for the present work. However, most took food crops as the research object and focused on using data envelopment analysis to explore single linear or nonlinear relationships between agricultural production scale and production efficiency and lacked an in-depth exploration of cash crops, especially in the apple industry. Moreover, previous studies have mostly only focused on basic statistical characteristics and knowledge cognition level [13, 16, 22, 24]. How does the planting scale of the apple

industry, as an important cash crop, affect production efficiency? What is the threshold for a moderate planting scale? These questions remain unanswered but are worthy of further study.

To overcome the limitations of the above mentioned literature, contributions of this paper are as follows: first, research to date has focused less on the relationship between planting scale and productivity in apple production, this paper from the perspective of planting scale, this study measured the efficiency level of the apple industry based on the Malmquist productivity index using DEA. On that basis, this study used a threshold regression model to analyze the characteristics of changes in the planting scale and efficiency of apples. Second, this study analyzes the TFP of apples and its contribution from the perspective of the dynamic fluctuation of time change. Thirdly, This study can be used to ensure the safety of agricultural production and explore optimal planting scale. It also provides a valuable reference for policy makers and practitioners to adjust or improve the development of the apple industry.

The structure of this paper is as follows: The second part introduces Materials and methods, the third part introduces data analysis and results, the fourth part introduces discussions, the last part introduces Conclusions and implications and limitations.

## Materials and methods

### Research area and data source

In the fruit industry in China, the apple occupies an important position. It is known as the "number one fruit." According to the China Agricultural Statistical Yearbook (2010–2018), there are seven major apple-producing provinces in China: Shaanxi, Shandong, Gansu, Henan, Shanxi, Hebei, and Liaoning. The apple production of Shandong Province ranks in second place after Shaanxi Province, reaching 9.522 million tons, accounting for 24.27% of the total production of the country, and is well worthy of the apple province. In addition, in Shandong Province, the Jiaodong Peninsula, Taiyi Mountain Area, and the North Shandong Plain are the main areas for apple cultivation. Among them, the Yantai apple is the most famous; in 2018, the Yantai apple had a regional brand value of 14.505 billion yuan and was the only brand valued at more than 100 million fruits.

Because of the limited availability of data, all data selected in this paper are made up of the above seven provinces from 2008–2018, and the apple production of these seven provinces accounts for more than 90% of the total production in China. All data are from the China Agricultural Statistical Yearbook, the National Agricultural Cost-Benefit Data Compilation, the website of the National Bureau of Statistics, and the China Agricultural Information Network.

### Measurement index

Based on the convention and the availability of data, this paper selects the yield of apple per hectare as the output variable and sets four input variables: land input, labor input, capital input, and fertilizer input. Land input refers to the planting area of the land. Labor input includes the domestic labor discount and hiring costs. Capital input refers to the cost of materials and services, including the cost of pesticides, fuel, and power. Fertilizer inputs include nitrogen fertilizer, phosphate fertilizer, potash fertilizer, compound fertilizer, and other fertilizer cost inputs. Table 1 shows the index system of the apple production efficiency measurement.

### Method selection and model construction

**Malmquist productivity index in the non-parametric form.** DEA is an evaluation method based on the relative effectiveness of input-output data. It is widely used in agricultural

**Table 1. Measuring index system of apple production efficiency.**

| Indicators | Variable | Variable declaration |
|---|---|---|
| **Input** | Land investment | Land size (ha) |
| | Labor input | Labor cost (10,000 yuan) |
| | Capital investment | Material and service costs (10,000 yuan) |
| | Fertilizer inputs | Fertilizer amount (10,000 Yuan) |
| **Output** | Apples per hectare | Production of apples per hectare (kilograms per hectare) |

economics and other fields [25–27]. The advantages of this method are as follows: First, it does not need to set a detailed function form, which avoids errors in model setting caused by subjective reasons [28, 29]. Second, other efficiency methods are generally limited to unit output and need to convert input and output into the same unit, while the DEA method does not need to unify the units of measurement of input variables, and the variable setting is more flexible. Third, the Malmquist productivity index method based on non-parametric DEA can reflect the changes in the efficiency of production units in different periods, which is more accurate and efficient [24, 30]. The method is mainly decomposed into technological progress (TECHCH) and technological efficiency change (EFFCH) for evaluation and analysis. The product of the two is the Malmquist productivity index, i.e., TFP = EFFCH×TECHCH. Based on the above analysis and using the existing research methods for reference [31], this study adopts the Malmquist productivity index method of data envelopment analysis (DEA) to calculate the production efficiency of the apple industry and to scientifically determine the influence of planting scale on production efficiency.

Malmquist index is defined by the distance function, and the Malmquist index based on output angle in the period of t is defined as:

$$M_0^t(x^t, y^t, x^{t+1}, y^{t+1}) = \frac{D_0^t(x^{t+1}, y^{t+1})}{D_0^t(x^t, y^t)}. \qquad (1)$$

Correspondingly, the Malmquist index of output angle in the period of T +1 is defined as:

$$M_0^{t+1}(x^t, y^t, x^{t+1}, y^{t+1}) = \frac{D_0^{t+1}(x^{t+1}, y^{t+1})}{D_0^{t+1}(x^t, y^t)}. \qquad (2)$$

Owing to the arbitrariness of period selection, in order to avoid the difference in this case, the geometric mean values of the Malmquist index in two different periods were selected, and the production technology in T and T +1 periods are referenced. The mathematical expression is as follows:

$$M_0(x^{t+1}, y^{t+1}, x^t, y^t) = \left[ \left( \frac{D_0^t(x^{t+1}, y^{t+1})}{D_0^t(x^t, y^t)} \right) \times \left( \frac{D_0^{t+1}(x^{t+1}, y^{t+1})}{D_0^{t+1}(x^t, y^t)} \right) \right]^{\frac{1}{2}}. \qquad (3)$$

Based on the calculation idea of the scholar Fare [30], under the condition of constant return to scale (CRS), that is, under the condition of limited output, the Malmquist productivity index successfully achieves the goal of maximum output, and in the process of empirical estimation, the maximum average productivity obtained is the best practice in the process of sample data measurement. In addition, the Malmquist productivity index can be decomposed into the technical progress index (TECH) and technological efficiency change (EFFCH), as shown

below:

$$M_0^t(x^t, y^t, x^{t+1}, y^{t+1}) = \left[\left(\frac{D_0^t(x^{t+1}, y^{t+1})}{D_0^t(x^t, y^t)}\right) \times \left(\frac{D_0^{t+1}(x^{t+1}, y^{t+1})}{D_0^{t+1}(x^t, y^t)}\right)\right]^{\frac{1}{2}}$$

$$= \frac{D_0^{t+1}(x^{t+1}, y^{t+1})}{D_0^t(x^t, y^t)} \left[\left(\frac{D_0^t(x^t, y^t)}{D_0^{t+1}(x^t, y^t)}\right) \times \left(\frac{D_0^t(x^{t+1}, y^{t+1})}{D_0^{t+1}(x^{t+1}, y^{t+1})}\right)\right]^{\frac{1}{2}} = \text{EFFCH} \times \text{TECH}$$

(4)

The index of technical efficiency change is $\text{EFFCH} = \dfrac{D_0^{t+1}(x^{t+1}, y^{t+1})}{D_0^t(x^t, y^t)}$. (5)

The index of technological progress efficiency is

$$\text{TECH} = \left[\left(\frac{D_0^t(x^t, y^t)}{D_0^{t+1}(x^t, y^t)}\right) \times \left(\frac{D_0^t(x^{t+1}, y^{t+1})}{D_0^{t+1}(x^{t+1}, y^{t+1})}\right)\right]^{\frac{1}{2}}$$

(6)

The above two formulas represent the changes in technical efficiency and the technological progress rate from T to T +1, respectively. If both of them are less than 1, the productivity will decline.

**Threshold regression model.** Studies have indicated that there may be a nonlinear relationship between production scale and efficiency [19]. The relationship between the threshold effect is when an economic parameter reaches a certain numerical threshold and causes other economic parameters to resort to other forms of development [31]. Therefore, on the basis of measuring the production efficiency of apples, to further clarify the relationship between the two, this article uses the Hansen (2000) [32] threshold regression model analysis of the nonlinear relationship between variables and further analysis of apple production scale and production efficiency.

The analysis model of this paper is set as follows:

$$\text{LnTFP}_{it} = C + \beta_1 \ln \text{applescale}_{it} \, I \, (q_{it} \leq \gamma) + \beta_2 \ln \text{applescale}_{it} \, I \, (q_{it} \geq \gamma) + \theta X_{it} + \varepsilon_{it},$$

Where $\text{TFP}_{it}$ is the total factor production efficiency, applescale$_{it}$ represents the average production scale of apples, and $X_{it}$ represents the control variables, including rural human capital, economic development level, and natural disasters. The affected area of apples in each province (1,000 hectares) is used to explain the natural disaster index, and the per-capita GDP index is used to explain the economic development level. Moreover, $\theta$ is the coefficient of all the control variables in the formula, and $\varepsilon_{it}$ is the random disturbance term.

## Statistical analysis

The production efficiency and scale of apples in different provinces are different in different years. To explore the specific influence and connection, this study uses DEAP2.1 software [33–36] to calculate the total factor production efficiency of seven provinces and cities from 2008–2018. In addition, when using the DEA method for measurement, the dynamic change in the total factor production efficiency of each province is analyzed, and the analysis method also further measures the technical progress and technical efficiency of each province. Therefore, through this change and efficiency situation, we can gain further insight into the change rule and trend of apple production efficiency [36], in which the total factor productivity index (expressed by TFP) in the main apple-producing areas is decomposed into the technological progress index (expressed by TECHCH) and technical efficiency index (expressed by EFFCH).

In addition, there is a nonlinear relationship between the apple planting scale and efficiency. In order to avoid the possible deviation of artificial interval division, this study uses the Hansen threshold regression model [32] to deeply analyze the nonlinear relationship between the apple planting scale and production efficiency. The specific steps are as follows: The first step is to determine whether there is a threshold effect; the second step confirms the specific threshold value and estimates the nonlinear equation based on it. Before estimating the threshold model, we first check the threshold value and then check the authenticity of the threshold value. In this study, Stata17.0 software is used to search the threshold values, from no threshold value to one threshold value, to three threshold values in turn. On this basis, the threshold model is estimated by econometrics. In this study, Bootstrap repeated sampling was performed 500 times to estimate the F statistics, P values, and critical values of different threshold numbers.

## Data analysis and results

### Apple total factor productivity growth and regional comparative analysis

Table 2 reveals the index values of TFP measurement and decomposition in seven major apple-producing areas in China from 2008 to 2018. Here, 1 is the cut-off point of apple production efficiency: A value greater than 1 indicates an increase in apple production efficiency; otherwise, it indicates a decline.

First, based on the apple-producing average TFP measurement and the decomposition of the indexes, apple-producing EFFCH since 1980 has increased to a certain extent; greater technical efficiency is the main driving force for the improvement of regional apple TFP. However, because the apple TECHCH there is backward, leading to TFP, there is a certain degree of deterioration. Since 2008, the average TFP in the seven provinces and cities selected in this paper fell by 6.40%. Moreover, the TFP of all provinces except for Henan (1.025) shows varying degrees of decline, and Shanxi's apple TFP shows a decline, the largest being at a rate of 12.10%. Furthermore, EFFCH shows an increasing trend on the whole, with an average annual growth rate of 0.6%, among which, except for Gansu (0.993) and Shanxi (0.996), EFFCH shows an increasing trend. The TECHCH rate shows a downward trend, with an average annual decline of 8%, and technological progress also shows a downward trend in the major regions.

From the perspective of specific interprovincial increase and decrease ratios, the difference between the provinces with the fastest growth rate of apple TFP (Henan, 1.9%) and the provinces with the largest decrease rate (Shanxi, -12.1) is 14%. The difference between the province with the fastest growth rate (Henan, 2.5%) and the province with the biggest decline rate (Gansu, -0.7%) is 3.2%. The difference between the provinces with the smallest decline in technological progress (0.6% in Henan) and those with the largest decline (11.7% in Shanxi) is

Table 2. Average value of TFP and its components in the seven main apple-producing areas from 2008–2018.

| Region | Total Factor Productivity Index (EFFCH) | Rate of Technological Progress (TECHCH) | Technical Efficiency (TFP) |
|---|---|---|---|
| Gansu | 0.993 | 0.908 | 0.902 |
| Hebei | 1.004 | 0.914 | 0.987 |
| Henan | 1.025 | 0.994 | 1.019 |
| Liaoning | 1.014 | 0.900 | 0.913 |
| Shandong | 1.010 | 0.916 | 0.925 |
| Shanxi | 0.996 | 0.883 | 0.879 |
| Shaanxi | 1.000 | 0.927 | 0.927 |

11.1%. It can be seen that there are significant differences in the apple TFP, apple TECHCH, and EFFCH in different provinces.

In order to comprehensively analyze the changes in apple production efficiency in these provinces and cities, explore its deep changes, this study analyzes the TFP of apples and its contribution from the perspective of the dynamic fluctuation of time change [37]. The changes in the time series are shown in Table 3.

The provinces of apple TFP in the cause of the degradation also exhibit some differences. There are some differences in five of the major apple-producing provinces (Hebei, Liaoning, Shandong, Henan, Shaanxi) where the technical efficiency (EFFCH) is greater than or equal to 1, indicating these provinces' technology efficiency (EFFCH) is attributed to the apple pull area TFP and progress of main power to a certain extent. However, because the apple TECHCH there is backward, leading to TFP, there is a certain degree of deterioration. Owing to the decline of both EFFCH and TECHCH in Gansu and Shanxi provinces, the double restriction leads to the decline of apple production efficiency. Since 2008, China's apple-producing technical progress index has been less than 1; the technological progress has decreased by the smallest percentage (0.6%) in Henan province, and the largest gap between provinces (Shanxi, 11.7%) is 11.1%. Poor instructions in apple industry technology progress are the main reason for the declining productivity of the apple industry.

Table 3 also reveals that the TFP of major apple-producing areas shows fluctuations to different degrees from 2008 to 2018. The TFP of Gansu shows a slightly fluctuating decline overall, which is consistent with the results in Table 2. Although slight growth is apparent in 2008, 2011, 2015, and 2017, and the total factor growth rate in most years is less than 1. In Hebei, technical efficiency drives technological progress. In Liaoning, EFFCH motivates TECHCH, and only technological progress (0.775) is a driver of technological efficiency (0.665) in 2014.

**Table 3. Time-series changes of apple total factor productivity and its contribution from 2008–2018.**

| Region | | 2008 | 2009 | 2010 | 2011 | 2012 | 2013 | 2014 | 2015 | 2016 | 2017 | 2018 |
|---|---|---|---|---|---|---|---|---|---|---|---|---|
| **Gansu** | EFFCH | 1.000 | 1.000 | 1.000 | 1.000 | 1.000 | 1.000 | 0.938 | 1.066 | 1.000 | 1.000 | 0.917 |
| | TECHCH | 1.166 | 0.727 | 0.732 | 1.083 | 0.849 | 0.847 | 0.833 | 1.091 | 0.833 | 1.008 | 0.819 |
| | TFP | 1.166 | 0.727 | 0.732 | 1.083 | 0.849 | 0.847 | 0.781 | 1.163 | 0.833 | 1.008 | 0.751 |
| **Hebei** | EFFCH | 1.000 | 0.875 | 0.925 | 1.082 | 0.895 | 1.169 | 1.081 | 1.010 | 1.000 | 1.000 | 1.000 |
| | TECHCH | 0.870 | 0.844 | 0.780 | 0.969 | 0.956 | 0.852 | 0.917 | 0.977 | 1.065 | 0.907 | 0.909 |
| | TFP | 0.870 | 0.739 | 0.722 | 1.048 | 0.856 | 0.996 | 0.991 | 0.987 | 1.065 | 0.907 | 0.909 |
| **Henan** | EFFCH | 1.000 | 1.000 | 1.000 | 1.000 | 1.000 | 1.000 | 1.000 | 1.000 | 1.000 | 1.000 | 1.277 |
| | TECHCH | 1.277 | 1.018 | 0.513 | 0.984 | 0.975 | 1.081 | 0.763 | 1.190 | 0.946 | 0.897 | 1.297 |
| | TFP | 1.277 | 1.018 | 0.513 | 0.984 | 0.975 | 1.081 | 0.763 | 1.190 | 0.946 | 0.897 | 1.656 |
| **Liaoning** | EFFCH | 1.000 | 1.000 | 1.000 | 1.000 | 1.000 | 1.000 | 0.665 | 1.445 | 0.909 | 1.144 | 0.986 |
| | TECHCH | 0.745 | 0.840 | 0.760 | 1.053 | 0.963 | 1.078 | 0.775 | 0.950 | 0.992 | 0.973 | 0.756 |
| | TFP | 0.745 | 0.840 | 0.760 | 1.053 | 0.963 | 1.078 | 0.515 | 1.373 | 0.902 | 1.113 | 0.745 |
| **Shandong** | EFFCH | 1.073 | 0.925 | 1.068 | 0.778 | 1.063 | 0.919 | 0.934 | 1.196 | 0.968 | 0.957 | 1.229 |
| | TECHCH | 0.949 | 0.843 | 0.827 | 0.941 | 0.931 | 0.835 | 0.923 | 0.925 | 1.000 | 0.945 | 0.815 |
| | TFP | 1.018 | 0.780 | 0.883 | 0.732 | 0.990 | 0.767 | 0.862 | 1.106 | 0.968 | 0.904 | 1.002 |
| **Shanxi** | EFFCH | 1.000 | 1.000 | 1.000 | 1.000 | 1.000 | 1.000 | 1.000 | 1.000 | 1.000 | 1.000 | 0.956 |
| | TECHCH | 0.944 | 0.717 | 0.865 | 0.956 | 0.789 | 0.876 | 0.925 | 0.918 | 1.123 | 0.989 | 0.608 |
| | TFP | 0.944 | 0.717 | 0.865 | 0.956 | 0.789 | 0.876 | 0.925 | 0.918 | 1.123 | 0.989 | 0.581 |
| **Shaanxi** | EFFCH | 1.000 | 1.000 | 1.000 | 1.000 | 1.000 | 1.000 | 1.000 | 1.000 | 1.000 | 1.000 | 1.000 |
| | TECHCH | 0.890 | 0.933 | 0.785 | 1.063 | 1.008 | 0.865 | 0.995 | 0.908 | 1.032 | 0.820 | 0.897 |
| | TFP | 0.890 | 0.933 | 0.785 | 1.063 | 1.008 | 0.865 | 0.995 | 0.908 | 1.032 | 0.820 | 0.897 |

The TFP of Henan, Shanxi, and Shaanxi is closely related to TECHCH and fluctuates with TECHCH. Shandong index of TFP is stable, but the overall growth rate is not high, especially as the lack of TECHCH hinders the EFFCH increase. Further analysis of Shandong's apple production resource allocation is unreasonable, as resource waste is an issue, resulting in a decline in apple quality and price, restricting the improvement of the overall technical efficiency of apple production, and affecting the development of the TFP.

## Threshold effect of apple planting scale efficiency

As shown in Table 4, the estimated value of the model with a single threshold is 7.515, at which time p = 0.002, indicating that the model with a single threshold is significant at the 1% significance level. The model values of the double threshold are 7.515 and 7.691, and P = 0.098 in this case, indicating that the double threshold model is significant at the 10% level. Moreover, P = 0.504 for the triple threshold model, indicating that the triple threshold model is not significant and does not pass the test. Therefore, we can judge the impact of the apple planting scale on apple production efficiency in each province and city, and there are two thresholds, 7.515 and 7.691, respectively. Different threshold numbers and their F statistics, corresponding probability P values, and critical values are obtained by the sampling of "Bootstrapping" 500 times. The results are shown in Table 4.

To judge whether the above threshold value is true, this study analyzes the model of the double threshold. Figs 1 and 2 are the likelihood ratio function diagrams, drawn according to the dual-threshold model. According to the likelihood ratio function diagram, the estimation of apple production efficiency threshold value and the construction of the confidence interval can be more intuitively understood, and then we can verify whether it is consistent with the real threshold value.

First, according to the results of double threshold regression and the corresponding likelihood ratio function graph analysis, we find that the double threshold values are 7.515 (see Fig 1) and 7.691 (see Fig 2), respectively. The 95% confidence interval of each threshold estimate is the critical value at the significance level of all LR < 5%, which corresponds to the interval below the dotted line in the figure. Second, to further analyze the impact of the apple planting scale on the apple production efficiency, the generalized panel regression model is expressed as follows:

$$y_{it} = \alpha_{it} + \beta_{it} x_{it} + \varepsilon_{it}.$$

Panel data reflect the influence of individuals and time, where $\alpha_{it}$ is a random variable, $\beta_{it}$ represents the coefficient of the explanatory variable, and $\varepsilon_{it}$ represents the disturbance term. The regression results of the panel threshold model are shown in Table 5 below.

Table 4. Threshold effect test of apple production scale.

| Model | F value | P value | BS degree | The critical value | | | Threshold value | 95% confidence interval |
|---|---|---|---|---|---|---|---|---|
| | | | | 1% | 5% | 10% | | |
| A single threshold | 21.681*** | 0.002 | 500 | 16.948 | 11.587 | 9.178 | 7.515 | [7.515,7.699] |
| Double threshold | 7.364* | 0.098 | 500 | 12.777 | 9.829 | 7.251 | 7.691 | [7.613,7.871] |
| Triple threshold | 0.608 | 0.504 | 500 | 2.961 | 2.051 | 1.649 | 7.955 | [7.942,7.986] |

Note: * indicates significance at the 10% level;

** indicates significance at the 5% level;

*** indicates significance at the 1% level

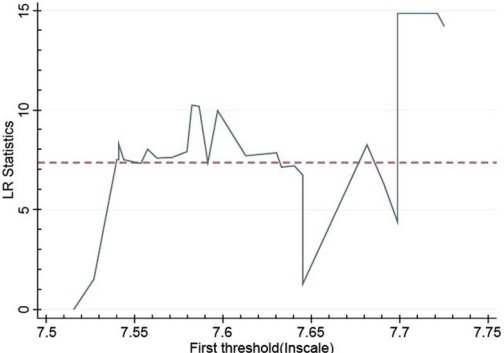

**Fig 1. Estimation and likelihood ratio function graph of the first threshold value produced by apple.**

The regression results of the panel threshold model are shown in Table 5. When the scale of labor production per worker is less than 7.515, its coefficient effect on apple planting efficiency is −1.66, which is significant at the level of 5%. This indicates that when the scale of apple planting is small, the apple production efficiency is less affected. When the planting scale of apples is higher than 7.515, the effect on apple production efficiency reaches 1.734, and it is significant at the level of 5%. When the production scale of apples exceeds 7.691, the planting efficiency of apples drops to 1.229, which is significant at the level of 10%. This indicates that the production scale of apples has a threshold effect on the production efficiency of apples and presents a trend from "negative effect to positive effect." In addition, the impact coefficient of rural human capital on apple production efficiency is 0.013, and it is significant at the level of 10%, indicating that the greater the rural human capital is, the greater the impact on apple production efficiency will be. Local economic development level has a similar effect on apple production efficiency. The higher the local economic development level is, the higher the apple production efficiency is [38]. The impact coefficient of the disaster index of natural disasters is −0.018, which has a negative correlation with the productivity of the apple industry.

## Discussions

Under the realistic dual constraints of the ecological environment and agricultural resources, agricultural development issues restrict the development of the global economy and society. Improving agricultural production efficiency has become the key to guaranteeing agricultural safety, increasing farmers' income and enhancing the international competitiveness of

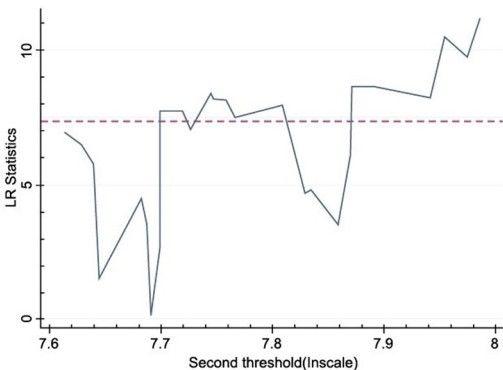

**Fig 2. Estimation and likelihood ratio function of apple's second production threshold.**

**Table 5. Regression results of the threshold model.**

| Variable | Coefficient | Standard deviation | P value |
|---|---|---|---|
| lnpergdp | 0.219*** | 0.073 | 0.002 |
| lnddisater | −0.018** | 0.021 | 0.033 |
| lnedu | 0.013* | 0.039 | 0.091 |
| lnapplescale1 | −1.660** | 0.820 | 0.104 |
| lnapplescale2 | 1.734** | 0.724 | 0.041 |
| lnapplescale3 | 1.229* | 0.351 | 0.072 |

Note: * indicates significant at 10% level;

** indicates significant at the 5% level;

***means significant at the 1% level; Lnapplscale1 means apple planting size < 7.515; 7.515 < lnapple scale2 means apple planting scale < 7.691; Lnapplescale3 represents apple planting scale BBB 0 7.515.

agricultural products. Moreover, moderate-scale operation is the inevitable trend of agricultural modernization [37, 39]. This study takes the apple industry as an example, calculates and analyzes the production efficiency of apples based on the Malmquist productivity index method of DEA, builds a threshold model based on the production efficiency measurement of apples, and studies the threshold effect of the scale of apple planting on the production efficiency of apples. In the existing literature, there have been few studies on the nonlinear effects of apple planting scale on efficiency. Based on the theory of the threshold model, this study focuses on the nonlinear effect of apple planting scale on efficiency. The results of this study confirm that there is a threshold effect of apple production scale on apple production efficiency—not that the bigger the planting scale is, the better it is. The matching of technological progress and land scale endowment has an impact on production efficiency [38, 40]. Some theoretical contributions have been made to improve the production efficiency of farmers and enable them to put forward new ideas but also for the relevant departments to improve and formulate relevant policies to provide new ideas.

The results of this study show that the analysis of TFP of apples in each main apple-producing area presents different development status. The technical efficiency of Hebei, Henan, and Shaanxi has been improved to a certain extent, but owing to the shortage of apple planting technology, the total factor production efficiency has deteriorated to a certain extent. One possible reason is that the aging of fruit farmers in Hebei Province, the structural shortage of labor, and the continuous rise of factor costs have become the bottleneck factors restricting the development of the apple industry. The labor shortage and factor cost rising trend is obvious, but the applicability of labor-saving technology is limited. In Liaoning, the reason is that old orchards are still using traditional production techniques, which have weakened the fruit trees, reduced the yield, and reduced the quality of the apples. The refrigeration equipment cannot keep fresh apples in paper bags to meet the needs of the high-end market and export market, and there is no cold chain transportation system. Moreover, the infrastructure of Liaoning Province is weak; the main reason may be that the development policy of the fruit industry is not stable, the regionalization of fruit trees is not scientific, some excellent varieties cannot be adjusted to local conditions, and there is blind planting, resulting in the occurrence of freezing damage, disease, a large area of fruit trees dying, and a great reduction in the productivity of apples. After 2011, the yield of Shandong was surpassed by that of Shaanxi, so the yield and planting area of apples declined. Some growers of old varieties with a low management level cut down trees or changed varieties for planting, which directly affected the production efficiency of apples. The planting period of the apple industry in Shandong Province is too

concentrated, and the pressure to pick and sell apples is great. Shaanxi and Shanbei are located in arid and semi-arid areas, belonging to the typical "rain-fed" agricultural areas. There are insufficient water source projects and a serious shortage of water diversion into apple orchards, which affect the yield and quality of apples.

The results also show that the production efficiency of apples in Gansu and Shanxi is degraded because of the double constraint of technical efficiency and technological progress. Because of the lag in the construction of the seed and seedling breeding system in Gansu, a large number of new orchards need to be imported from Shandong, Shaanxi, and other provinces, and the variety of fruit trees is mixed, so the quality of new orchards is not high. The extension of cultivation technology is not timely, and the technical training and technical mastery are not synchronized. To ensure the quality of varieties, some farmers re-graft all the stubble in the second year of planting, and the fruit trees often do not bear fruit for several years, increasing production costs. Shanxi apple's commercialization processing level is low. Moreover, the packaging is simple, the maturity is inconsistent, the coloring is not neat, there is a lack of competitiveness in the fruit market, the product's added value is low, and the processing technology is deficient.

In addition, based on an analysis of the threshold effect, the study findings show that although there is a threshold effect of apple production scale on apple production efficiency, it is not true that the larger the business scale is, the better. In other words, production efficiency and scale can develop on a certain scale. According to one study, seven Chinese apple-producing areas can undergo moderate-scale cultivation [34, 35], namely land circulation dispersal, fragmentation of land circulation to apple operation at large, professional cooperatives, and other ways to realize centralized management. All of these can produce the scale effect and improve the efficiency of apple production. Nationally, there is a lot of room to save land costs, labor costs, and fertilizer inputs while maintaining the same yield per mu.

As for the control variables, the coefficients of rural human capital and economic development level are 0.013 and 0.219, respectively, indicating that the impact on apple production efficiency is positive. The impact coefficient of the disaster index of natural disasters is −0.018, which has a negative correlation with the productivity of apples. Some studies indicate that the larger the rural human capital is, the more obvious the effect on the improvement of apple production efficiency [31, 32, 41]. Therefore, the rural population can be trained to improve the human capital level of rural labor and thereby improve apple production efficiency. In addition, the higher the level of local economic development, the higher the efficiency of apple production, so economic development should drive the synchronous improvement of apple production. At the same time, the impact of natural disasters on apple productivity is negatively correlated. Provinces should respond to the impact of natural disasters with technology and improve risk awareness and the ability to cope with disasters. The above findings are consistent with the existing research that "apple production scale has a positive relationship with apple production efficiency," but do not suggest that bigger plants are better.

## Conclusions and implications and limitations

This study aims to accurately grasp the planting scale and production efficiency of China's apple industry and understand the competitiveness of the apple industry in different regions to promote farmers' income increase. In this study, we used the Malmquist productivity index of the DEA method to measure the apple industrial efficiency in the seven main apple-producing areas of China and then used the threshold model to analyze the nonlinear characteristics of the change in apple production scale and efficiency in China. The main conclusions of this study are as follows:

First, in the seven provinces and cities selected for this study, except for Henan, TFP showed different degrees of decline. Average TFP showed a decrease of 6.40%, and the TFP of apples in Shanxi showed the greatest decline.

Second, we further analyzed the specific contributing factors of each index of apple TFP decomposition, the results of apple TFP, and the contribution degree from the perspective of time-series change. We found that the TFP of the main apple-producing areas fluctuated to different degrees from 2008 to 2018. The TFP of different regions was greatly different, and the technological progress and technical efficiency were different in different years, leading to obvious differences in the TFP of different provinces. For example, the lack of TECHCH in Gansu, Hebei, Liaoning, Shanxi, and Shaanxi offset the contribution rate of EFFCH, resulting in negative growth in the TFP. EFFCH in Henan promoted the positive growth of TFP. Another example is that in Shandong Province, as the driving force of apple production mainly depends on the improvement of technical efficiency, regional technological progress has a poor influence on production efficiency. This also reveals that apple production in different provinces is affected by many factors, such as the natural endowment of scale operations and local economic development.

Finally, apple production efficiency in different regions of China showed regional differences and time-series fluctuations. The apple planting scale had a threshold effect, and the impact on apple production efficiency showed a "negative effect–positive effect" trend. Specifically, when the scale of apple planting does not exceed the threshold value of 7.515, the scale hurts apple production efficiency. When the apple planting scale is between 7.515 and 7.691, the effect of the apple planting scale on apple production efficiency becomes a positive effect. When the scale of apple planting exceeds the second threshold value, the effect of scale on apple production efficiency becomes smaller. The impact of rural human capital and local economic development level on apple production efficiency is positively correlated while the impact of natural disasters on apple productivity is negatively correlated.

The conclusions of this study provide some important practical implications.

(1) Effective matching with apple planting scale: Government departments can reduce the threshold for apple growers, apple industry cooperative organizations, and new apple growers to use new agricultural technologies if they adopt relevant supportive policies, promote the diffusion and dissemination of technological progress in the apple industry, improve technical efficiency, and optimize the allocation of input resources. (2) Improving the ability of apple production to cope with risks. Due to the frequent occurrence of global climate change and extreme weather, drought, hail, freezing, disease, and insect and other natural and biological disasters occur from time to time. Moreover, orchard drainage and irrigation capacity are insufficient. Therefore, the government should strengthen agricultural infrastructure construction and provide technical guidance services and key support in apple-growing infrastructure construction to improve agricultural comprehensive development capacity. (3) Guiding moderate-scale apple production. At present, China's apple production is mainly scattered among individual farmers. Farmers generally have a low level of knowledge and culture; they can only rely on years of planting experience to plant fruit trees. Therefore, the government innovates the way of land transfer, reduces the cost of land transfer, establishes relevant laws and policies supporting land transfer, and healthily develops the moderate-scale cultivation of the apple industry. (4) Playing a positive role in apple planting. Through professional skills training, the human capital of local apple growers can be improved, and the sustainable development of the apple industry can be promoted fundamentally. In addition, the state should guide large apple producers to play a leading role and guide advanced factors such as technology, capital, and talent to gather in large-scale production areas to form a reasonable industrial layout and scale. (5) Raising the level of Apple's industrial operation. First, through apple information sharing,

we can increase the promotion of apple production technology. Strengthening the functions of apple industry monitoring and apple market risk management is conducive to overcoming the contradictions between farmers and the market. Second, by relying on "Internet +," we can expand apple's sales channels and realize the integration of production and sales.(6)Government departments should actively promote the confirmation of farmland rights, cultivate the farmland leasing and rural loan markets, and set up perfect relevant laws to satisfy large-scale agricultural production.

Although this paper has carried on comprehensive research, limitations exist. First, in the analysis of the threshold effect of apple planting scale and apple production efficiency, the influences of natural factors such as land quality, quantity, and plot area were not considered in the variable selection. The production efficiency of apples is affected by various intermediate variables and control variables. Therefore, in future research, some necessary natural conditions and other control variables can be included in the model for further investigation. Second, in future studies, energy input will be considered for inclusion in the theoretical analysis framework, and agricultural non-point source pollution and agricultural carbon emissions will be regarded as non-expected outputs to conduct a relevant analysis, which is also the content of the next step of research.

## Supporting information

**S1 Data.**
(XLSX)

## Acknowledgments

The first author of this article would like to thank the academic committee of the University for its support to my scientific research work.

## Author Contributions

**Conceptualization:** Yu Sun, Yonghua Lu.

**Data curation:** Yu Sun, Yonghua Lu.

**Formal analysis:** Yu Sun.

**Methodology:** Yu Sun.

**Writing – original draft:** Yu Sun.

**Writing – review & editing:** Yu Sun, Zichun Wang, Mingyue Li.

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
