## [Decision Letter · Decision Letter 0]

15 May 2021

PONE-D-21-13599

Production Efficiency and Change Characteristics of China’s Apple Industry in Terms of Planting Scale

PLOS ONE

Dear Dr. Yu Sun,

Thank you for submitting your manuscript to PLOS ONE. After careful consideration, we feel that it has merit but does not fully meet PLOS ONE’s publication criteria as it currently stands. Therefore, we invite you to submit a revised version of the manuscript that addresses the points raised during the review process.

We look forward to receiving your revised manuscript.

Kind regards,

László VASA, PhD

Academic Editor

PLOS ONE

Journal Requirements:

[Author Contributions: Conceptualization, Y.S.; methodology, Y.S. and L.Y.; data curation, Y.S.; writing—original draft preparation, Y.S. and Z.W.; editing, Y.S and L.M.There is no financial competition for this article.].

5. Thank you for stating the following in the Funding Section of your manuscript:

[This research was funded by the Shandong Province Modern Agricultural Industrial Technology System (SDAIT-01-13), the Project of Shandong Provincial Department of Education (No: M2020111), and the** high****-level talent research start-up fund of Qingdao Agricultural University (No: ****663 / 1116710)**.]

 [This research was funded by the Shandong Province Modern Agricultural Industrial Technology System (SDAIT-01-13), the Project of Shandong Provincial Department of Education (No: M2020111)]

6. Please amend the manuscript submission data (via Edit Submission) to include author Yonghua Lu.

Reviewers' comments:

Reviewer's Responses to Questions

**Comments to the Author**

1. Is the manuscript technically sound, and do the data support the conclusions?

Reviewer #1: Partly

Reviewer #2: Yes

Reviewer #3: Yes

2. Has the statistical analysis been performed appropriately and rigorously? 

Reviewer #1: Yes

Reviewer #2: Yes

Reviewer #3: Yes

3. Have the authors made all data underlying the findings in their manuscript fully available?

Reviewer #1: Yes

Reviewer #2: Yes

Reviewer #3: Yes

4. Is the manuscript presented in an intelligible fashion and written in standard English?

Reviewer #1: Yes

Reviewer #2: Yes

Reviewer #3: Yes

5. Review Comments to the Author

Reviewer #1: The paper analyses a quite interesting and important topic of apple production of China and its productivity parameters based on the available statistical datasets, with the appropriate methodology. The results, as well as the conclusions are correct and based on the metodology. Conclusions are right and also limitations are indicated which fact is more tan welcomed.

Howevere, for agreeing with publication of this paper, I suggest:

- The keywords should be more focused and precise (e.g. indicating China; considering apple production instead of apple industry).

- No formal and separated literature review in the paper can be found. I recommend to do one, in a critical, analytical and comprehensive way where the essential international sources are processed.

After these improvements, I can accept this paper for publishing.

Reviewer #2: The authors need to more accurately take into account the peculiarities of apple production (biological, temporal, climatic, organizational, technological, etc.) and include in the article the statistical data used in the calculations.

Reviewer #3: The submitted manuscript meets the requirements for scientific articles. The authors asked research questions.

What is the impact of planting scale in the apple industry on production efficiency, as an important cost-intensive factor? What is the threshold for a moderate planting scale?

The article presents the measurement of the apple industry performance level on the basis of the Malmquist productivity index using DEA. In the study, a threshold model based on the measurement of apple production efficiency was built, which examines the threshold effect of the apple planting scale on apple production efficiency.

Previous studies have focused less on the relationship between the scale of planting and apple productivity.

There has been little research in the existing literature on the non-linear effects of the apple planting scale on yield. Based on the threshold model theory, this study focuses on the non-linear effect of the apple planting scale on yield. The results of this study confirm that there is a threshold impact of the apple production scale on production efficiency - this does not mean that the larger the planting scale, the better.

The authors reviewed the literature and indicated that the improvement in production efficiency is the result of many factors - family relationships, apple planting experience, natural environmental factors (e.g. climate change, geographic location and natural disasters), equipment and management style (e.g. pesticide use) and irrigation, marketing and organization).

All these are important factors that determine that the expansion of the production scale affects the improvement of production efficiency.

The authors investigated the impact of planting scale in the apple industry.

They identified important factors that determine that a change in the scale of production improves its efficiency.

The research used data on seven provinces from statistical yearbooks. The authors report that some of the missing data are based on calculations from the yearly data.

I suggest clarification in section 2.1. Research area and data source studies, which have missing data based on calculations from other year data.

I recommend the article for publication.

6. PLOS authors have the option to publish the peer review history of their article (what does this mean?). If published, this will include your full peer review and any attached files.

Reviewer #1: No

Reviewer #2: **Yes: **Dr. K.A. Zhichkin

Reviewer #3: **Yes: **Aleksandra Łakomiak

---

## [Author Response · Author response to Decision Letter 0]

27 Jun 2021

Reviewer #1:

Thank you very much. We have updated our list of keywords, replaced the “apple industry” with “apple production,” and reworked the original introduction into a separate section as a literature review to render our manuscript clearer.

Thank you for your valuable advice.

Reviewer #2:

Thank you very much. Paper data from China’s official data “The National Agricultural Cost-Benefit Data Compilation,” “The China Agricultural Statistical Yearbook,” and “The National Agricultural Cost-Benefit Data Compilation” in the capital investment costs including technology. There are no climate factors, or factors such as time, organization, and technology. This issue requires a lot of field research. Future research will pay more attention to this.Some necessary natural conditions and other variables can be included in the model for further investigation.

 Thank you for your valuable advice.

Reviewer #3: 

Thank you very much for your valuable advice. 

First, as mentioned in the conclusion, the specific contents are as follows. In this study, we used the Malmquist productivity index in the context of the DEA method to assess the industrial efficiency of the seven main apple-producing areas of China and then used a threshold model to analyze the nonlinear characteristics of the change in apple production scale and efficiency in China. The influence of apple planting scale on production efficiency is as follows:

Apple production efficiency in different regions of China showed regional differences and time-series fluctuations. The apple planting scale had a threshold effect, and the impact on apple production efficiency showed a "negative effect–positive effect" trend. Specifically, when the scale of apple planting does not exceed the threshold value of 7.515, the scale hurts apple production efficiency. When the apple planting scale is between 7.515 and 7.691, the effect of the apple planting scale on apple production efficiency becomes a positive effect. When the scale of apple planting exceeds the second threshold value, the effect of scale on apple production efficiency becomes smaller. The impact of rural human capital and local economic development level on apple production efficiency is positively correlated while the impact of natural disasters on apple productivity is negatively correlated.

I regret that my choice of expression caused misunderstanding. All data in this paper are from China Agricultural Statistical Yearbook, the National Agricultural Cost-Benefit Data Compilation, the website of the National Bureau of Statistics, and the China Agricultural Information Network. Other data not included in the statistics have not been used in this paper. The data in this paper are not based on data extrapolated from other years.I adjusted and deleted the ambiguous sentence in the original.

National Bureau of Statistics: http://www.stats.gov.cn/

China Agricultural Information Network: http://www.agri.cn/

Thank you for your valuable advice.

---

## [Editor Report · Decision Letter 1]

5 Jul 2021

Production Efficiency and Change Characteristics of China’s Apple Industry in Terms of Planting Scale

PONE-D-21-13599R1

Dear Dr. Yonghua Lu,

We’re pleased to inform you that your manuscript has been judged scientifically suitable for publication and will be formally accepted for publication once it meets all outstanding technical requirements.

Kind regards,

László Vasa, PhD

Academic Editor

PLOS ONE
---

## [Editor Report · Acceptance letter]

13 Jul 2021

PONE-D-21-13599R1 

Production Efficiency and Change Characteristics of China’s Apple Industry in Terms of Planting Scale 

Dear Dr. Lu:

I'm pleased to inform you that your manuscript has been deemed suitable for publication in PLOS ONE. Congratulations! Your manuscript is now with our production department. 

Kind regards, 

on behalf of

Prof. Dr. László Vasa 

Academic Editor

PLOS ONE